# Prognostic factors associated with failure of total elbow replacement: a protocol for a systematic review

Zaid Hamoodi [1,2] Celina K Gehringer,[1,3] Lucy M Bull,[4] Tom Hughes [5]
Lianne Kearsley-Fleet,[1] Jamie C Sergeant [1,3] Adam C Watts[2]

LK-F, JCS and ACW contributed equally.

LK-F, JCS and ACW are joint senior authors.

For numbered affiliations see end of article.

**Correspondence to**
Dr Zaid Hamoodi;
zaid.hamoodi-2@postgrad.manchester.ac.uk

## ABSTRACT

**Introduction** Total elbow replacement (TER) has higher failure rates requiring revision surgery compared with the replacement of other joints. Understanding the factors associated with failure is essential for informed decision-making between patients and clinicians, and for reducing the failure rate. This review aims to identify, describe and appraise the literature examining prognostic factors for failure of TER.

**Methods and analysis** This systematic review will be conducted and reported in line with the Preferred Reporting Items for Systematic Review and Meta-Analysis Protocols guidelines. Electronic literature searches will be conducted using Medline, EMBASE, PubMed and Cochrane. The search strategy will be broad, including a combination of subject headings (MESH) and free text search. This search will be supplemented with a screening of reference lists of the included studies and relevant reviews. Two independent reviewers will screen all search results in two stages (title and abstract, and full text) based on the Population, Index prognostic factor, Comparator prognostic factor, Outcome, Time and Setting criteria. The types of evidence included will be randomised trials, non-randomised trials, prospective and retrospective cohort studies, registry studies and case–control studies. If the literature lacks enough studies, then case series with 50 or more TERs will be considered for inclusion. Data extraction and risk of bias assessment for included studies will be performed by two independent reviewers using the Checklist for Critical Appraisal and Data Extraction for Systematic Reviews of Prediction Modelling Studies for Prognostic Factors and Quality In Prognostic Studies tools.

Meta-analyses of prognostic estimates for each factor will be undertaken for studies that are deemed to be sufficiently robust and comparable. Several challenges are likely to arise due to heterogeneity between studies, therefore, subgroup and sensitivity analyses will be performed to account for the differences between studies. Heterogeneity will be assessed using Q and $I^2$ statistics. If $I^2 > 40\%$ then pooled estimates will not be reported. When quantitative synthesis is not possible, a narrative synthesis will be undertaken. The quality of the evidence for each prognostic factor will be assessed using the Grades of Recommendation Assessment, Development and Evaluation tool.

**PROSPERO registration number** CRD42023384756.

## STRENGTHS AND LIMITATIONS OF THIS STUDY

⇒ This is the first systematic review to focus on identifying and appraising studies evaluating prognostic factors associated with failure of total elbow replacement.

⇒ This review will follow the methodological advances in performing systematic reviews and meta-analyses in prognosis research proposed by the PROGnosis REsearch Strategy and The Cochrane Prognosis Methods Group.

⇒ This review will use the recommended checklists and tools including the PICOTS model (Population, Index prognostic factor, Comparator prognostic factor, Outcome, Time and Setting) for study selection, the CHARMS-PF (Checklist for Critical Appraisal and Data Extraction for Systematic Reviews of Prediction Modelling Studies for Prognostic Factors) for data extraction, and the QUIPS tool (Quality In Prognostic Studies) to assess the risk of bias.

⇒ This review may be limited by the quality of and the heterogeneity between the included studies. We aim to address this by performing statistical analysis of heterogeneity combined with subgroup and sensitivity analyses.

## INTRODUCTION

Total elbow replacement (TER) is a recognised treatment for painful arthritic elbow and while rheumatoid arthritis remains the main indication, it is increasingly performed for acute trauma and post-traumatic arthritis.[1] Despite technological advances, TER has higher failure rates than other total joint replacements, with a reported 5-year failure rate of 6%–10% and 10-year failure rate of 15%–19% compared with a 3% 5-year failure rate and <5% 10-year failure rate in total hip replacement.[1–8] In the event of TER failure, surgical revision is usually required but this procedure carries a risk of adverse events such as ongoing pain, disruption of the elbow extensor mechanism, reduction of bone stock, infection, ulnar nerve injury and poor function.[9] Understanding the factors associated with failure is essential to inform the



development of strategies designed to reduce failure risk and the need for subsequent revision surgery, which is a burden for both patients and society.[10]

A prognostic factor is defined as any variable that is associated with a risk of a health outcome among people with a particular health condition, in this context the outcome is failure, and the condition is patients with TER.[11 12] Several existing studies have examined possible factors that may be associated with the risk of failure in TER, including primary pathology, prosthesis type, surgeon's experience, implant fixation and whether surgery is performed in a specialised centre.[2 13–19] However, some of the findings have been inconsistent. For example, some studies examining the association between the underlying pathology and TER failure found rheumatoid arthritis to be associated with a worse prognosis while others found post-traumatic arthritis to be associated with a worse prognosis.[6 16]

To the best of our knowledge, there are no systematic reviews available that have appraised, pooled and synthesised all existing studies relating to prognostic factors for TER failure, which means that the quality of the current evidence base is unknown. This also means that it is unclear whether potential factors have been derived from single exploratory studies (thus having provisional prognostic value) or replicated and confirmed in multiple studies (thus demonstrating robust prognostic value).

Therefore, the purpose of this systematic review is to identify, describe, appraise and synthesise all the current literature examining the prognostic factors associated with failure of TER. A better understanding of prognostic factors may help to explain the differences in risk of failure between patients,[20] which may facilitate decisions about whether or not to proceed with TER as a definitive management option. An improved understanding of prognostic factors may also help reduce the number of potential TER revisions by improving treatment selection[21] and may help to identify areas for novel interventions.[11] This review will also identify gaps in the evidence base to aid the planning of future research and could potentially pave the way to developing a prognostic model in TER that could be used to make individualised patient predictions.

## METHODS AND ANALYSIS

A systematic review will be conducted and reported in line with the Preferred Reporting Items for Systematic Review and Meta-Analysis Protocols guidelines.[22] It has been registered at the International Prospective Register of Systematic Reviews and will follow the published guidance and recommendation produced by the PROGnosis REsearch Strategy and The Cochrane Prognosis Methods Group on how to undertake a systematic review and meta-analysis in prognostic factors research.[11 12 20 23–25]

### Patients and public involvement

The study aim was codesigned with the Wrightington Hospital Patients and Public Involvement group, including members with long-term health conditions and experience of joint replacement surgery. The group also contributed to a list of possible prognostic factors to be included in the review and their ongoing involvement will help ensure that the review is relevant to patients.

### Study eligibility

All peer-reviewed studies will be included if prognostic factors are investigated in patients who have undergone a primary TER, and if the selection criteria in table 1 are met (based on the Population, Index prognostic factor, Comparator prognostic factor, Outcome, Time and Setting (PICOTS) model).[12] Studies must include a human population and provide a prognostic estimate evaluating prognostic factors for failure of TER. This initial list of candidate prognostic factors to be considered is listed in table 2 but, importantly, this is not exhaustive and studies will also be included if other candidate prognostic factors have been investigated. Although the primary interest is in prognostic factors measured before surgery, we will not exclude prognostic factors measured intraoperatively or postoperatively.

The outcome of failure will be defined as revision surgery. The definition of revision surgery varies between

| Table 1 | PICOTS used as selection criteria |
|---|---|
| Population | Any human population with total elbow replacement |
| Index prognostic factor | All possible prognostic factors |
| Comparator | Not applicable because all prognostic factors will be systematically identified, and the evidence will be summarised |
| Outcome | The outcome is failure which is defined as revision surgery. Revision surgery is defined as any secondary surgery to the prosthesis, this includes addition, removal or alterations to all or part of the construct on the same elbow occurring at any time following the primary total elbow replacement. |
| Timing | Follow-up at any time after the initial TER will be evaluated (all time periods) |
| Setting | Any healthcare setting |

PICOTS, Population, Index prognostic factor, Comparator prognostic factor, Outcome, Timing, Setting.; TER, total elbow replacement.

**Table 2** Possible prognostic factors investigated in the literature

| | |
|---|---|
| Patient factors | Age<br>Comorbidities or American Society of Anaesthesiologists status<br>Ethnicity<br>Hand dominance<br>Indication for surgery<br>Sex/gender<br>Socioeconomic status<br>Weight or body mass index<br>Occupation<br>Activity/functional level |
| Implant factors | Fixation type<br>Implant brand<br>Implant design (linked/unlinked) |
| Surgery factors | Length of surgery<br>Surgical approach<br>Surgical technique (eg, cementation technique)<br>Use of venous thromboembolism prophylaxis<br>Use of antibiotics |
| Surgeons/ hospital factors | Hospital experience (numbers per year)<br>Surgeon's experience (numbers per year)<br>Surgeon's grade |

joint registries and is likely to vary between other non-registry studies,[26] therefore, a broad definition for revision surgery will be used which is any secondary surgery to the prosthesis, this includes addition, removal or alterations to all or part of the construct on the same elbow occurring at any time following the primary TER. This may lead to the inclusion of studies that evaluate non-revision secondary surgeries as part of a composite outcome. If this occurs, these studies will undergo subgroup/sensitivity analyses.

The types of evidence included will be randomised trials, non-randomised trials, prospective and retrospective cohort studies, registry studies and case–control studies. If the literature is lacking in studies of these types, then a case series with 50 TERs or more will be considered for inclusion. Results from case series will be interpreted with caution due to the higher risk of bias (RoB), especially selection bias. Review articles, surveys, case reports and conference abstracts will be excluded.

If the literature is published in a non-English language, an English version will be identified, or the author contacted to check if an English version is available. If an English version is unavailable, the study will be excluded. A list of excluded non-English language studies will be summarised in online supplemental material.

Studies that report only laboratory/biomechanical work, methodology for identifying new prognostic factors, or results from animal studies will be excluded. Furthermore, if multiple papers are published using the same or overlapping datasets, then only one of these papers will be included, which will be based on the largest number

of patients, the most detailed results and/or the longest follow-up time.

## Search strategy

A broad search will be conducted to ensure all the relevant studies are captured in the search (high sensitivity). As the reporting of prognostic factor studies is generally poor,[27] the search will aim to capture all published studies for TER/arthroplasty (low specificity).

Electronic searches will be conducted using OVID Medline, Embase, PubMed and Cochrane Library databases. The search strategy will include a combination of subject headings (MESH) and free text searches (online supplemental file 1). The proposed search terms were developed with guidance from an experienced information scientist at the University of Manchester. The electronic search will not be limited by the date of publication or the language, or to the human population. This search will be supplemented with a screening of reference lists of the included studies and relevant review studies to identify any further studies that could have been missed in the electronic search. All duplicates will be removed using EndNote V.20 (Philadelphia, USA).

## Study screening

The study screening process will undergo two phases: (1) screening of titles and abstracts and (2) full-text screening. The screening process will be blinded and all studies in each phase will be screened independently by two reviewers against the specified inclusion and exclusion criteria (table 1). ZH will screen 100% of the studies and the second independent review will be performed by CKG and LMB (50% each) (figure 1). The screening will be conservative, and studies will be included in the next round of screening unless there is an agreement to remove them. Copies of the relevant and unclear studies will be obtained and read thoroughly by the two independent reviewers. Any disagreement will be resolved by discussion or involvement of a third reviewer (JCS). The reason to exclude any study in the full-text screening phase will be provided in online supplemental file. Rayyan software (Cambridge, USA) will be used for the management of the search results.

Pilot testing of the selection process will be undertaken using 30 articles selected at random and screened by the reviewers. A meeting will be held to review the screening process to determine whether any changes need to be made.

## Data extraction

The data will be extracted using a standardised tool that will be based on the Checklist for Critical Appraisal and Data Extraction for Systematic Reviews of Prediction Modelling Studies for Prognostic Factors (CHARMS-PF).[12] CHARMS-PF covers nine domains: (1) source of data, (2) participants, (3) outcomes to be predicted, (4) prognostic factors, (5) sample size, (6) missing data, (7) analysis, (8) results and (9) interpretation and discussion.

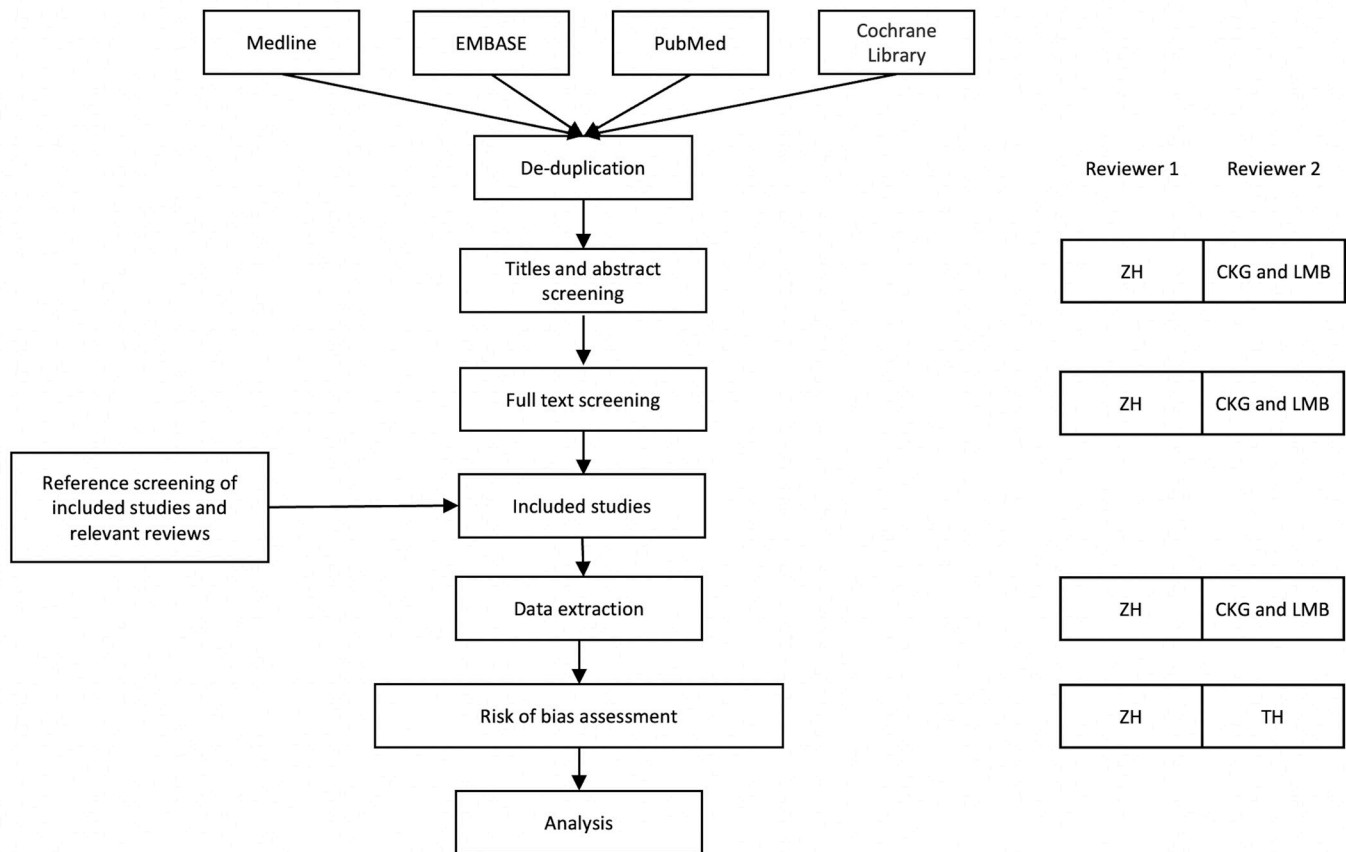

**Figure 1** Summary of the systematic review process.

The data from all included studies will be extracted by two independent reviewers, ZH will complete 100% of data extraction and the second independent data extraction will be performed by CKG and LMB (50% each), and any disagreement will be resolved by discussion. Where disagreement remains, a third reviewer (JCS) will be asked to assess the data and the majority opinion will be accepted. The data extraction tool will initially undergo pilot testing using 10% of the included studies chosen at random and a meeting will be held to review and determine whether any changes need to be made.

### Risk of bias

The methodological quality or RoB in the included studies will be assessed using the Quality In Prognostic Studies tool which has been recommended and developed specifically for prognostic factor studies[28 29]. The tool has six domains: (1) selection of study participation, (2) study attrition, (3) prognostic factor measurement, (4) outcome measurement, (5) study covariates and (6) statistical analysis and reporting.

Two independent reviewers, (ZH) and (TH), will assess each domain and mark with responses 'yes', 'no' or 'unclear'. The RoB in each domain will then be classified into 'low', 'moderate' or 'high'. Any disagreement between the two reviewers will be resolved by discussion. Where there is disagreement, a third reviewer (JCS) will

be asked to assess the study and the majority opinion will be accepted.

### Data synthesis and meta-analysis

For the quantitative synthesis, prognostic estimates and their precisions will be extracted for each prognostic factor. This will include the risk ratio (RR), OR or HR to measure the prognostic effect, and SEs and CIs to measure the precision. Mean difference will also be accepted to measure associations for continuous variables. Both adjusted and unadjusted prognostic effects will be considered; however, adjusted effect measures will be favoured in the interpretations.[12]

Meta-analyses for each prognostic factor will be undertaken for studies that are deemed to be sufficiently robust and comparable. Several challenges are likely to arise due to heterogeneity between studies; therefore, subgroup analysis will be undertaken of groups of studies with similar characteristics. If the number of studies allows, then separate meta-analyses will be undertaken for groups with similar prognostic effect measures, for example:

► HRs, ORs and RRs separately.
► Unadjusted and adjusted associations separately.
► Prognostic factor effects at distinct cut points (or groups of similar cut points) separately.
► Prognostic factor effects corresponding to a linear trend (association) separately.

► Prognostic factor effects corresponding to non-linear trends separately.
► Prognostic factors at different times of measurement (preoperatively, intraoperatively or postoperatively) separately.
► Each method of measurement (for factors and outcomes) separately.

Sensitivity analysis will also be performed when there is heterogeneity between studies caused by differences in:

► The outcome (revision surgery) definition.
► The participations baseline characteristics (casemix).
► The RoB.
► The study designs.
► The type of TER included in the study.

If heterogeneity persists despite subgroup analyses, then a random effect approach will be used to allow for unexplained heterogeneity across included studies. This will be performed if five or more studies are included in a meta-analysis, otherwise, a fixed effect approach will be used.[30]

Statistical analyses will be performed using R V.4.2.2 or later and RStudio software (Vienna, Austria). If quantitative synthesis is not possible because of issues such as a paucity of evidence, overall evidence quality and methodological heterogeneity, a narrative synthesis of the evidence will be undertaken.

## Assessment of heterogeneity

Pooled data will be used in the review and heterogeneity will be assessed between each group using forest plots with 95% CIs for the pooled measures. Heterogeneity will also be assessed by calculating the Q statistic and $I^2$. If there is heterogeneity across the included studies ($I^2 > 40\%$),[31] then the overall pooled data estimate will not be reported. The heterogeneity results will still be reported.

## Reporting bias

Reporting deficiencies will be assessed using a funnel plot which addresses 'small-study effects'. The funnel plot will be used when 10 or more studies are examining the same prognostic factor and included in the same meta-analysis.[32] The Peters' and Debray's tests will be used to test for asymmetry in the funnel plot.[32 33] The asymmetry in the plot cannot differentiate between differences caused by bias (publication or selective reporting biases) or heterogeneity; therefore, the results will be interpreted with caution.

## Reporting

The quality of evidence relating to each prognostic factor will be evaluated using the adapted Grades of Recommendation Assessment, Development and Evaluation (GRADE) framework that is specific to systematic reviews of prognostic factor research.[34] The GRADE contains five domains:

► RoB.
► Inconsistency.
► Imprecision.

► Indirectness.
► And publication bias.

The GRADE system will be used to summarise whether the evidence for each prognostic value is of high, moderate, low or very low quality. The online The GRADEpro Guideline Development Tool will be used.

**Author affiliations**
[1]Centre for Epidemiology Versus Arthritis, Centre for Musculoskeletal Research, University of Manchester, Manchester Academic Health Science Centre, Manchester, UK
[2]Upper Limb Unit, Wrightington Wigan and Leigh NHS Foundation Trust, Wigan, UK
[3]Centre for Biostatistics, School of Health Sciences, Faculty of Biology, Medicine and Health, Manchester Academic Health Science Centre, The University of Manchester, Manchester, UK
[4]Technology Department, Health Navigator Ltd, London, UK
[5]Department of Health Professions, Manchester Metropolitan University, Manchester, UK

**Contributors** ZH, LK-F, ACW and JCS provided the idea of the topic. ZH designed and wrote the protocol. CKG and LMB contributed to designing the search strategy, screening process and data extraction. TH contributed to the methodology of the risk of bias assessment and group for sensitivity analysis. JCS contributed to the design of the statistical methods. ACW contributed to a list of a priori prognostic factors. ZH coordinated the whole process. All authors read, provided feedback, input into the methodology and approved the final manuscript. LK-F, ACW and JCS contributed to this work equally.

**Funding** This work was supported by funding from: The joint National Joint Registry (NJR) and Royal College of Surgeons of England (RCS England) Research Fellowship (grant number: NA). The John Charnley Trust (grant number: NA). Versus Arthritis (grant number: 21755). ZH research fellowship is funded by the NJR/ RCSEng and his MD degree is funnded by the John Charnley Trust.

**Competing interests** None declared.

**Patient and public involvement** Patients and/or the public were involved in the design, or conduct, or reporting, or dissemination plans of this research. Refer to the Methods section for further details.

**Patient consent for publication** Not applicable.

**Provenance and peer review** Not commissioned; externally peer reviewed.

**ORCID iDs**
Zaid Hamoodi http://orcid.org/0000-0003-0045-7257
Tom Hughes http://orcid.org/0000-0003-2266-6615
Jamie C Sergeant http://orcid.org/0000-0002-9000-4413

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
