## [Reviewer comments · BMJ Open]

ARTICLE DETAILS

TITLE (PROVISIONAL)	Prognostic factors associated with failure of total elbow replacement: a protocol for a systematic review
AUTHORS	Hamoodi, Zaid; Gehringer, Celina; Bull, Lucy; Hughes, Tom; Kearsley-Fleet, Lianne; Sergeant, Jamie; Watts, Adam

VERSION 1 – REVIEW

REVIEWER	Heifner, John J. Miami Orthopaedic Research Foundation
REVIEW RETURNED	27-Mar-2023

GENERAL COMMENTS	This abstract/proposal is well-written, clear and concise and represents an important topic that would add the the current literature.
--

REVIEWER	Lambert, Simon University College London Hospitals NHS Foundation Trust, Department of Trauma and Orthopaedic Surgery
REVIEW RETURNED	17-Jan-2023

GENERAL COMMENTS	This is a well-written, well-argued, concise but comprehensive account of the protocol for a proposed review of factors resulting in failure of total elbow replacements. The rationale for the study has been clearly described. The study is relevant in developing our current understanding of the outcomes of total elbow replacement. The purpose of the study - to improve knowledge of methods to increase the reliability and durability of total elbow replacement - has been clearly defined. The review is constructed accurately: the methodology has been described clearly, and in accordance with current recommendations and guidance. The proposed statistical analyses are pertinent and the possibility of inadequate data to submit for analysis accounted for. The authors will use a narrative assessment if the data in review are of poor quality. I support this submission with no recommendations for revision.
---

VERSION 1 – AUTHOR RESPONSE

We have made 2 revisions to the original manuscript. When we started to conduct the review, it was decided to only include studies that directly report prognostic estimates. This is because the indirect methods would only provide unadjusted prognostic estimates with low precision^{1,2}. It was also operationally more straightforward to use the availability of a direct prognostic estimate as an inclusion criterion. Hence, we've updated the protocol to reflect this.

Those changes are highlighted within the manuscript. Please see below the changes. All page numbers refer to the revised manuscript file with tracked changes.

Change 1:

Page 6-7/Lines 157-158 : The statement “provide a quantitative result or give tabulated individual patient data (IPD)” changed to “provide a prognostic estimate”.

Change 2:

Page 10/Lines 261-264: The paragraph “For missing estimates of prognostic values, the primary author will be contacted and estimates will be requested, or, where appropriate, statistical methods described by Parmar et al (Stat Med 1998) and Tierney et al (Trials 2007) will be used to estimate unadjusted hazard ratios and their variances[30 31]” was deleted.

References:

- 1) Tierney, J.F., Stewart, L.A., Ghersi, D. *et al.* Practical methods for incorporating summary time-to-event data into meta-analysis. *Trials* **8**, 16 (2007). <https://doi.org/10.1186/1745-6215-8-16>
- 2) Riley R D, Moons K G M, Snell K I E, Ensor J, Hooft L, Altman D G et al. A guide to systematic review and meta-analysis of prognostic factor studies *BMJ* 2019; 364 :k4597 doi:10.1136/bmj.k4597